



# Estimation of westward auroral electrojet current with magnetometer chain data

Marina A. Evdokimova and Anatoli A. Petrukovich

Space Research Institute of the Russian Academy of Sciences, Moscow, 117997, Russia

**Correspondence:** M. A. Evdokimova (evdokimari@mail.ru)

**Abstract.** We investigate one-dimensional models of westward substorm electrojet, using magnetic field observations along a meridian chain. We review two linear models of Kotikov et al. (1987) and Popov et al. (2001) with the large number of elementary currents at fixed positions. They can be applied to a magnetometer chain with many magnetic stations. A new nonlinear method with one current element is designed for the cases with small number of stations. We illustrate performance

of these methods using data from IMAGE and Yamal Peninsula stations. Several corrective measures are proposed to account for unphysical solutions or local extrema of the optimized functions. We also advertize a generic maximum likelyhood approach to a problem, usable for any empiric model.

## 1 Introduction

A ground-based magnetometer is the oldest instrument for space weather research. Data from hundreds of permanent and temporary magnetic stations all over the world are available. With magnetic records one can study the evolution of the main geomagnetic field, as well as geomagnetic variations. The most of the latter are driven by the magnetospheric and ionospheric currents, ultimately depending on solar activity. In particular, magnetic records are used to characterize the strength of geomagnetic substorms. The main substorm characteristic is amplitude of magnetic variations in the northern auroral zone, summarized

with AE/AU/AL geomagnetic indices. These variations are driven primarily by the westward auroral electrojet, which is an electric current, short-cutting the magnetotail cross-tail current (Ganushkina et al., 2018).

The goal of a dozen of AE/AU/AL stations is to catch the global maximum of a magnetic perturbation at all longitudes. To study electrojet and substorm dynamics in detail one needs to track at least one meridional profile of auroral geomagnetic variations with a local north-south chain of stations. The most famous and accessible are Scandinavian IMAGE chain (Viljanen

and Hakkinen, 1997), and Canada/Alaska chains. Meridional electrojet profile depends on substorm phase and strength of solar wind driving. In the course of a substorm, the activity zone first shifts equatorwards during growth phase, then, after an onset, it retreats polewards. For stronger substorms auroral zone shifts equatorwards.



While the primary measured parameter is magnetic field, it needs to be converted to electric current, which then can be compared with magnetospheric currents and used to quantify the substorm as a plasma phenomenon. Alternatively, one can compute geoelectric field, affecting pipelines or electric powerlines. The ionospheric parameters in the auroral zone, such as electron density and conductivity, are also of interest (Untiedt and Baumjohann, 1993).

5    A number of quantitative and semi-quantitative approaches were developed to convert from magnetic field to electric current in the auroral zone. A 2D-model of equivalent ionospheric currents can be implemented, if stations are distributed both along longitude and latitude (Amm and Viljanen, 1999). Several 1D algorithms are also available. Kotikov et al. (1987) approximated an electrojet with the series of current wires evenly distributed at 100-km altitude. Popov et al. (2001) introduced electrojet as a set of current strips of a finite width at 115-km altitude. These models are described in detail in the next section. With a simpler approach the Norwegian station network was used to define boundaries of the auroral oval, tracking maxima of vertical magnetic component (Johnsen, 2013). Kamide et al. (1982) suggested a simple method to estimate electric current density with one station only (given in Appendix). With a statistical approach, average oval boundaries can be related with $AL$ index (Starkov, 1994; Vorobjev et al., 2013). The Starkov (1994) model is provided in Appendix. Note, however, that almost all oval models return the boundaries of auroral lights or precipitations, rather than the boundaries of auroral currents. There exist also more global models, recovering electric currents from a distributed set of stations (e.g., Mishin , 1990)

The most of these methods, using instantaneous measurements, require a large number of stations to discover the electrojet spatial structure. However in many local time sectors the station network is sparse. In this report we develop the simple model of westward electrojet and the relevant solution scheme, which can be used with small number of stations (in fact, even with 2–3). We also describe some other useful algorithms. The key specifics of our approach is essential use of the vertical component of geomagnetic field ($Z$).

For the illustration we use two typical substorms with the sudden onsets and clear negative bays, gradually moving northward (Figure 1). The first case was registered 24 November, 1996 by the IMAGE network and was widely studied elsewhere (Petrukovich, 1999; Raeder et al., 2001). The second one was recorded at the Yamal peninsusla (Papitashvili et al., 1985). Time resolution of data is 1 minute. The station coordinates are in Table 1.





**Table 1.** List of geomagnetic coordinates of magnetometers

| Station | Lat. | Long. |
|---------|------|-------|
| NAL | 75.25 | 112.08 |
| HOR | 74.13 | 109.59 |
| HOP | 73.06 | 115.10 |
| SOR | 67.34 | 106.17 |
| TRO | 66.64 | 102.90 |
| KEV | 66.32 | 109.24 |
| MAS | 66.18 | 106.42 |
| KIL | 65.94 | 103.80 |
| KIR | 64.69 | 102.64 |
| SOD | 63.92 | 107.26 |
| PEL | 63.55 | 104.92 |
| OUJ | 60.99 | 106.14 |
| NUR | 56.89 | 102.18 |
| BEY | 68.18 | 146.87 |
| KHS | 66.19 | 143.21 |
| SKD | 61.82 | 141.50 |

## 2   Solution algorithms

### 2.1   General approach

We use the following approximation of the one-dimensional westward auroral electrojet (Figure 2): (1) electrojet flows at a fixed altitude of 110 km above the flat land; (2) electrojet is infinitely thin vertically; (3) electrojet flows along the latitude; (4) electrojet does not vary with longitude.

Magnetic disturbances in question are deviations from the quiet field, which has to be subtracted from the measurements. To determine the quiet level, we average magnetic data of 5 quietest days of the month, when the substorm occurred (Chapman and Bartels (1940)). The model latitude range spans $\pm 4$ degrees from the south-most and north-most stations (for the models with many elementary currents). The input magnetic field disturbance is forced to be zero at the edges of this range, to avoid nonphysical solutions. The ground magnetic disturbances are produced by the ionospheric current (electrojet) and the corresponding induction current inside Earth. The model latitudinal profile of ionospheric current is reconstructed using the



**Figure 1.** Examples of IMAGE (left) and Yamal (right) substorms

north-south $X$ and vertical $Z$ magnetic components measured at some set of ground observatories (magnetic stations). At the moment we ignore the $Y$ component of magnetic field.

## 2.2 Separation of external and internal field components

Ground magnetic disturbances can be described as:

$$X = X_e + X_i, \qquad Z = Z_e + Z_i, \tag{1}$$





where indices "$e$" and "$i$" denote external and internal components. According to Pudovkin (1960) difference between external and internal components at any point $x$ along meridian can be calculated as (here $H$ is horizontal field component):

$$H_e(x) - H_i(x) = -\frac{1}{\pi} \int_{-\infty}^{\infty} \frac{Z(\xi)}{\xi - x} d\xi,$$

$$Z_e(x) - Z_i(x) = \frac{1}{\pi} \int_{-\infty}^{\infty} \frac{H(\xi)}{\xi - x} d\xi. \tag{2}$$

So, external field components are:

$$H_e(x) = \frac{1}{2} \left[ H(x) + Int_H(x) \right],$$

$$Z_e(x) = \frac{1}{2} \left[ Z(x) + Int_Z(x) \right],$$

$$Int_H(x) = -\frac{1}{\pi} \int_{-\infty}^{\infty} \frac{Z(\xi)}{\xi - x} d\xi,$$

$$Int_Z(x) = \frac{1}{\pi} \int_{-\infty}^{\infty} \frac{H(\xi)}{\xi - x} d\xi. \tag{3}$$

This method works well in a case of a dense magnetometer chain with the large number of stations. $H(\xi)$ and $Z(\xi)$ are obtained with the linear or spline interpolation of the measured magnetic disturbance (forced to zero at the edges of the modelled latitude range, see previous subsection). Integrals are calculated over the same latitude range.

For the magnetometer chains with small number of stations we have to use the simpler method (Petrov, 1982) with the constant empirically justified coefficients:

$$X_e = \frac{2}{3} \cdot X, \qquad Z_e = 1 \cdot Z. \tag{4}$$

### 2.3    Solution scheme

We formulate the general maximum likelihood estimation (MLE) solution. We choose the model parameters, maximizing the likelihood function $L$.

$$L = \left[ \prod_{k=1}^{N} P_k \left( X_k, Z_k, Model(\overrightarrow{p}) \right) \right] \times P_P(\overrightarrow{p}) \tag{5}$$

Here $N$ — number of stations, $X_k$ and $Z_k$ — disturbance of the magnetic field, caused by the electrojet current, measured at the station $k$ (with the background field and induction field subtracted). $P_k$ — probability to observe given magnetic fields $X_k$ and $Z_k$ for some electrojet model with the parameter vector $\overrightarrow{p}$. $P_P$ — some apriori probabilities for $\overrightarrow{p}$.

Apriori information (aka priors) may be predictions from the statistical models or some common sense limitations, such as flatness of the spatial profile. The latter variant is also known as regularization. Regularization might be technically necessary





for the under-determined problems, when the number of free parameters is larger, than the number of degrees of freedom in the sample (number of the independent measurements).

In this investigation we use one of the most simple MLE variants, assuming Gaussian distribution of the model residuals, and solving the general OLS inverse problem.

$$-2lnL = \sum_{k=1}^{N} \left[ \frac{1}{\sigma_X^2} \left( \delta X_k - \delta X_{km_n}(\overrightarrow{p}) \right)^2 + \frac{1}{\sigma_Z^2} \left( \delta Z_k - \delta Z_{km_n}(\overrightarrow{p}) \right)^2 \right] + Q_r, \quad (6)$$

Here $X_{km_n}$ and $Z_{km_n}$ — calculated model disturbances, $Q_r$ denotes possible additional constraints. $\sigma_X$ ($\sigma_Z$) are standard variations of the measured $X$ ($Z$) components (at all used stations at a given time).

The parameter vector is determined looking for the minimum of $-2lnL$. If the whole model is linear with respect to the

parameter vector $\overrightarrow{p}$, the standard matrix inversion technique is applied for the solution. The nonlinear variants are solved here with the Levenberg-Marquardt algorithm, implemented in Matlab. This method needs specification of some initial values of the model parameters, and is then moving along the gradient of the optimization function towards the minimum. Unlike linear regression, such methods for nonlinear problems do not guarantee the unique solution due to existence of local minima, etc.

The errors of the model parameters $\overrightarrow{p}$, are calculated as inverted Hessian of $\ln L$:

$$cov(\overrightarrow{p}) = \left( \frac{\partial^2 \ln L}{\partial p_i \partial p_j} \right)^{-1}, \quad (7)$$

.

## 2.4  Model 1

The first described model was suggested by Kotikov et al. (1987). It includes the large number of the infinitely thin, fixed wires with the unknown currents. Wires are evenly distributed within the modeled latitude range, $\pm 4^o$ from the equator-most and the

pole-most stations. Magnetic field at the edge wires is set to zero.

$$\delta X_{km_1} = \frac{\mu_0 h}{2\pi} \sum_{j=1}^{M} \frac{I_j}{h^2 + \Delta x_{jk}^2},$$

$$\delta Z_{km_1} = \frac{\mu_0}{2\pi} \sum_{j=1}^{M} \frac{I_j \Delta x_{jk}}{h^2 + \Delta x_{jk}^2} \quad (8)$$

Here $h$ — height of the wires, $M$ — number of wires, $I_j$ — currents, $j = 1...M$, $\Delta x_{jk} = x_j - x_k$ — difference in coordinates of the wire $j$ and station $k$ along the magnetic meridian. The model magnetic disturbances $\delta X_{km_1}$ and $\delta Z_{km_1}$ depend on the

unknown model parameters $I_j$ linearly.

Regularization, suggested by the authors, is:

$$Q_r = \alpha \sum_{j=1}^{M} (I_j - I_{aj})^2 + q \sum_{j=2}^{M} (I_j - I_{j-1})^2, \quad (9)$$





**Figure 2.** The model scheme

where $I_{aj}$ — current at the previous time step. Coefficient $\alpha$ doesn't allow currents to change too fast (controls smoothness in the time domain), $q$ controls smoothness of the current profile along the meridian. Regularization is necessary, since the number of wires (of the model parameters) can be larger, than the number of stations (50 wires are proposed in the original paper). Still, the number of stations should be large enough (e.g., like in IMAGE chain), to provide enough information on the

5    spatial inhomogeneity of the current.





## 2.5 Model 2

The second described model was suggested by Popov et al. (2001). It is fundamentally similar to Model 1, except it consists of the evenly distributed strips with the unknown current density.

$$\delta X_{km_2} = \frac{\mu_0}{2\pi} \sum_{i=1}^{M} j_i \left( \arctan \frac{\Delta x_{ik} + d}{h} - \arctan \frac{\Delta x_{ik} - d}{h} \right),$$

$$\delta Z_{km_2} = \frac{\mu_0}{4\pi} \sum_{i=1}^{M} j_i \ln \frac{h^2 + (\Delta x_{ik} + d)^2}{h^2 + (\Delta x_{ik} - d)^2} \tag{10}$$

$$\tag{11}$$

where $d$ — half-width of the strip, $\Delta x_{jk} = x_j - x_k$ — difference in coordinates of the strip center $j$ and station $k$. Positions of the strips are fixed. Disturbances $\delta X_{km_2}$ and $\delta Z_{km_2}$ depend on the unknown model parameters $j_i$ linearly.

Regularization, suggested by the authors, is:

$$Q_r = q \sum_{i=2}^{M} (j_i - j_{i-1})^2 + \beta \sum_{i=2}^{M} j_i{}^2, \tag{12}$$

Here coefficient $q$ is responsible for smoothness of the current profile along the latitude, while $\beta$ limits the maximal current amplitude. Regularization is necessary, since large number of the strips is propozed in the original paper.

## 2.6 Model 3

For a small number of stations one needs a simpler model with one element of the electric current. The Model 1 is inconvenient, since a single infinitely thin current will return the unphysical magnetic profile. We use a version of Model 2, with one current strip with floating borders. The optimal unknown model parameters are: the current density, the low-latitude and high-latitude electrojet boundaries (explained in the following section). This model is nonlinear.

$$\delta X_{km_3} = \frac{\mu_0}{2\pi} j \left( \arctan \frac{x_k - x_l}{h} - \arctan \frac{x_k - x_h}{h} \right),$$

$$\delta Z_{km_3} = \frac{\mu_0}{4\pi} j \ln \frac{h^2 + (x_k - x_h)^2}{h^2 + (x_k - x_l)^2} \tag{13}$$

where $j$ — current density in a strip, $x_h, x_l$ — coordinates of the high-latitude and the low-latitude current borders respectively, $x_k$ – coordinate of the station $k$.

## 3 Model tests and algorithm adjustments

### 3.1 Number of wires and regularization

In Model 1 each infinitely thin wire creates a characteristic spatial peak of magnetic field with the latitudinal scale approximately equal to the height of the wire. Since height ($\sim 100$ km or $\sim 1^o$ of latitude) is much smaller, than the typical electrojet





width and the modelled latitude domain, a small set of wires will generate an unphysical magnetic profile with the several sharp minima (for westward electrojet). Fig. 3 (left panel) presents such Model 1 runs, using the Example 1 with 8 and 15 wires (with no regularization). Both variants return oscillating magnetic profiles, indicating that thr number of wires is insufficient. Note, that the case with 15 wires exhibits also another problem, typical for the models with too many parameters: some wires are

attributed with positive currents, creating positive excursions of magnetic field between stations, which are not supported with any evidence (measured field).

The linear model with 15 wires becomes underdetermined, since the number of independent inputs (the double number of stations) is comparable or smaller, than the number of unknowns. The underdetermined solution usually results in physically unrealistic large and very variable values of (here) elementary currents ideally cancelling each other at the magnetic stations,

where measurements are available (Fig. 3, right panel, model with 50 wires, red curves).

To ensure the suffiently flat electrojet profile one needs a denser current network with the separation much smaller than the height, but too sharp variations between stations need to be damped. The standard way to solve this problem is to use the so-called regularization procedure, penalizing variability and/or amplitude of the model parameters. Introduction of the regularization term in Model 1 with some reasonable coefficient $q \sim 1$ effectively reduces unwanted variations of currents, still

preserving reasonable complexity of the latitudinal profile (Figure 3, right, magenta curve, in comparison with blue and red ones).

Here, with Figure 3, it is important to note several aspects, related with the applicability of such 1D models. First, Model 1 reasonably well reconstructs $X$ component: empty circles in Figure 3 (right panel) always correspond to the measured data (black stars). However, the flattened Model 1 (with regularization) often fails to reproduce extreme $Z$ values (such as at latitude

$75^{o}$).

Secondly, when the station coverage is sparse (for IMAGE in the Norwegian/Barents sea, with the stations only at the mainland and Svalbard), even the model with the sufficient regularization may return the positive currents (above $75^{o}$, purple curve). This positive current results in the positive model $X$ values right in the gaps between stations. However all available stations measure only negative $X$, thus presence of positive current has no direct confirmation. These issues are further elaborated in

Discussion. With many elementary electric currents, it is possible to describe a relatively complex spatial profile of an electrojet, without the need to explicitly define the nonlinear latitudinal profile. Elementary currents can be placed at some evenly spaced fixed positions, the only free model parameters are electric current amplitudes in the numerator of the functional form (Eq. 8), so the model remains linear. The spatial inhomogeneity of an electrojet is well described by these changing amplitudes.

### 3.2   Selection of parameters of nonlinear model

The most natural variant for a case with the small number of stations is to use one strip from Model 2. Then the free parameters are current density, center and half-width of the strip. However, this variant has several drawbacks.

Current density and width of the electroject are strongly anticorrelated. Almost same magnetic field can be produced with a variety of current strips with the differing width and current density, but the same total current. Correlation of parameters





**Figure 3.** Event 24-Nov-1996, 23:09:00 UT. Left: Model 1 with 8 and 15 wires with no regularization. Right: Model 1 with 50 wires without and with the regularization. Measured field is shown with black stars, model field and current — with open circles.

complicates the error analysis, since standard error bars are produced by the diagonal elements of the error matrix (Eq. 7). Correlation of the parameters creates large nondiagonal elements, which often avoid sufficient attention.

The second drawback is related with the definition of electrojet boundaries. For example, if there is no station in the relevant position to catch a poleward boundary, the corresponding error will be propagated to both parameters: electrojet center and

5  width.



Thus the optimal Model 3 has three parameters: current density, poleward and equatorward boundaries. All parameters are defined almost independently. The current density mostly depends on the largest observed $X$ component disturbance, the boundaries — on the sign of $Z$ component at the nearest station.

When the number of stations is small, they might be quite often located not optimally relative to a specific electrojet. To illustrate how this problem is handled with Model 3 we resurrect one latitude profile from Example 1 (Figure 4). Fig. 4 (left) shows the model for the case with all stations, while the right panel shows two variants. The red curve corresponds to the case with three stations, two southward and one northward of electrojet, and the model electrojet is identical to that in the right panel (only the current density error is larger). However the case with four stations (blue curve), all equatorward of the electrojet, results in a substantially different model with the shifted poleward border. This border is also defined with a substantially larger error. To get this particular solution one needs also to avoid local minimum, this issue is described in the next subsection.

### 3.3 Avoiding local minima

Unlike with the linear regression, determination of the right non-linear solution is not guaranteed. All algorithms are sequential and may lead to local, rather than the global minimum of the target function (Eqs. 5, 6). The result may depend on the initial approximation of the model parameters, which needs to be specified to start the search. There are several standard ways to avoid local minima in a more or less automatic way.

The first approach is to introduce a prior — some apriori information on location of electrojet boundaries or electrojet amplitude. The apriori boundaries can be taken, e.g., from the Starkov model (Starkov, 1994) (Appendix A). As an input for Starkov (1994) model one can take either $AL$ index or local maximal negative $X$ component (from the modelled magnetic chain data). Then one may define in Eq. 6 $Q_r = w_d(d - d_0)^2$, where $d$ is some parameter, $d_0$ is apriori value, $w_d$ is some weight. This form penalizes any strong deviations from the apriori value. Thinking about a solution process as a descent along the local gradient in some landscape of the minimized function, introduction of a prior modifies this landscape, removing the local minima. However, though effective in some cases, this approach turned out to be very sensitive to selection of weights, which have to be specified manually for each model run.

The second approach is to use a so-called multistart algorithm. We generate a normally randomized set of initial conditions around a Starkov model solution, run Model 3 several times, and choose a result with the minimal residuals (Eq. 6). For the case of Fig. 4 (right panel, blue curve) we show the map of 50 initial conditions (for the boundary locations only). Starkov model is shown with red point. The solutions, starting from the filled black circles lead to the absolute minimum, shown by filled blue point. The empty black circles lead to the local minima (blue open circles). Since Model 3 is computationally simple, the method works well, and it is not necessary to fill densely the parameter space during the randomization.

### 3.4 Model 3 test and false global minimum problem

We illustrate Model 3 operation, running it for the whole Event 2 (Figure 6). On the left panel the time profiles of magnetic field, current density and electrojet boundaries are shown. This was rather strong substorm with the negative bay almost –1000 nT. Generally Model 3 returns reasonable results for magnetic profiles (Fig. 6a), but the electrojet boundaries are somewhat





**Figure 4.** Event 24-Nov-1996, 23:09:00 UT. Effect of the station selection. Left — many stations on both sides of the electrojet. Right — small number of stations on both sides of electrojet (red) and on one side only (blue). Error bars (standard deviation) are shown with the thin lines in panels (a) and (d).

different from the statistical Starkov model (Fig. 6c). During the growth phase (1600-1645 UT) the real electrojet is more poleward, which may be related with absence of a station at a sufficiently southward location. During the extended recovery phase (after 1800 UT) the electrojet is consistently more southward. The detailed analysis of this substorm, however, is beyond the goals of this report.

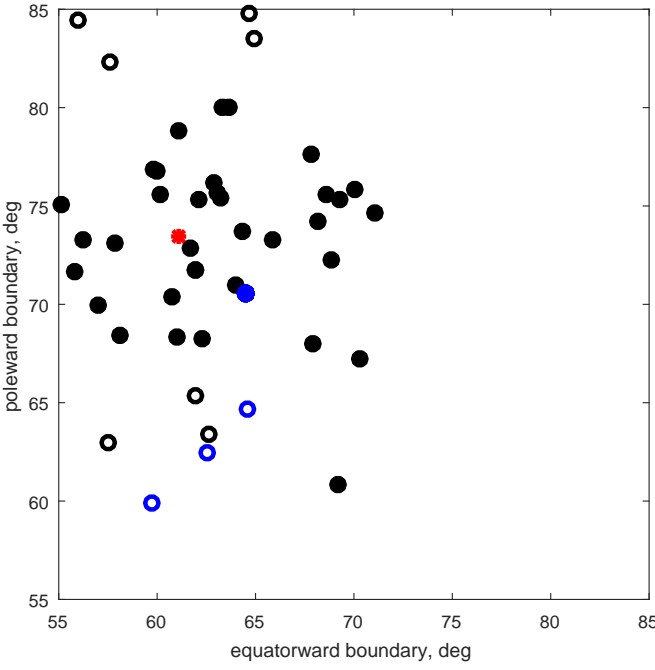

**Figure 5.** Event 24-Nov-1996, 23:09:00 UT. The map of initial conditions and final electrojet boundaries for the case in Fig. 5, right, blue curve. Initial conditions are shown with black circles. Starkov model (a center of randomised initials) is shown with red point. Absolute minimum is shown with blue filled circle. Local minima — with blue open circles. See text for further details.

Besides these easily interpretable results, at some moments the model reports definitely unphysical electrojet parameters, appearing in Fig. 6b,c as spikes. We highlight the four time instants with the problems of various kinds (shown with color vertical lines). The detailed model results for these instants are shown on the right panels of Figure 6.

The black vertical line (at 18:15:30 UT, Fig 6 left) and corresponding black curves (right panels) show fully reliable result
5   with small errors. The blue lines and curves for 17:37:30 UT report the case with the unreliable poleward border, which is even above $90^o$. Here all three stations fall on more equatorward side of electrojet (all $Z$ values are negative). The uncertainty interval for the poleward border is very large and extends down to a very reasonable latitude of $75^o$. The corresponding equatorward border and current density are well defined, as expected. A similar error, but for the equatorward border, occurs for 19:50:30 UT (red color). Here all three stations have positive $Z$.

10   More serious problem arises, if all three model parameters are physically incorrect as for 17:55:30 UT (magenta). Here the model returns electrojet with the zero width and very high current density. The model magnetic field profile shows very narrow dip between the stations with the amplitude 1.5 times larger than the really observed field. The model current density is very large and is out of scale.

The described problems are features of the true global maximum in the mathematical solution and can not be resolved within
15   the core model algorithm. They have to be removed with some additional physical considerations. In a case with one unreliable





border, one can fix the troubled parameter at some limiting values, e.g. $55^o$ and $85^o$, but anyway these numbers are not justified by any observations.

Somewhat counterintuitively, the situation is simpler for the case of the infinitely thin electrojet. One can force the current density to be equal to Kamide et al. (1982) estimate (see Appendix B). Then the model returns more reasonable, but still rather narrow (two degrees wide) electrojet (Figure 6, magenta dashed line). Magenta model in Fig. 6d corresponds to this adjusted solution. A substantial $X$ value at the equator-most station at $62^o$ still suggests that the real electrojet is wider, than the result, but the solution here balances both $X$ and $Z$ residuals.

## 4 Final algorithm for Model 3 with small number of stations

The optimal method to compute electrojet parameters with Model 3 and small number of stations is summarized below.

1. Select substorm interval of interest, preferably with the clear westward electrojet.

2. Subtract quiet magnetic field.

3. Subtract internal component of magnetic field using constant coefficients (Eq. 4).

4. Repeat following actions for all time instants with 1-min or 5-min cadence.

5. Create a set of initial latitudes normally distributed around boundaries of Starkov (1994). Initial current density can be taken equal to Kamide et al. (1982) estimate or also randomized.

6. For each set of initial conditions solve the minimization problem (Eqs. 6,7,12). Solution with the smallest residuals is final.

7. Check values of parameters and errors to determine reliability of individual parameters. If necessary, repeat computation of the reduced model with the fixed current density, using Kamide et al. (1982) estimate.

## 5 Discussion

The proposed 1D algorithms are computationally simple and efficiently recover auroral electrojet parameters in configurations like that of westward electrojet, developing during substorm expansion phase. Possibility to use only few magnetic stations substantially increase a span of longitudes, at which such modeling is possible. Determined electrojet amplitude and location can be used for a variety of studies, including, for example, comparison of electrojet boundaries with the oval boundaries, comparison of electrojet amplitude with that registered in space using AMPERE project data (Anderson et al., 2000), or with magnetospheric modeling. A potentially interesting is to develop with the Supermag dataset (Gjerloev, 2009) some extended auroral electrojet index, including electrojet total strength and location. Finally the developed technique can be used to recover storm-time electrojets, which move to lower latitudes with the sparser station coverage.





To be fully confident in the reconstructed meridional profile of the electrojet, one needs the station set dense enough at all latitudes in question. A five-degree gap of the IMAGE chain in the ocean appears often too large for such a model. The one-degree step, approximately equal to the electrojet height, is definitely sufficient. Assuming additionally some minimal electrojet width (e.g., two degrees), one can allow the equivalent couple-degree step. To capture only three electrojet parameters

(magnitude and borders, Model 3), the stations need to be somewhat offset on both sides with respect to actual electrojet location.

The described models have some natural physical limitations. First of all, any deviations from 1D are effectively averaged out. Some issues, such as deflection from latitudinal direction, can be handled with the reasonable complication of the model (including $Y$ component in consideration). The Model 3 can be also modified to use some bell-shaped electrojet profile. This

variant may potentially decrease effects of unphysically sharp electrojet edges. It is reasonable also to increase averaging, switching to 5-min step.

It should be specially noted, that analysis of our test data reveals frequent apparent inconsistency between $X$ and $Z$ magnetic components in 1D approximation. Visually it can be identified with "too large" $Z$ excursions, comparable with the expected $X$ values. In a gap with station location, Models 1 and 2, taking into account such $Z$ values, may generate unreasonable electrojet

latitude profiles, including reverse currents, which are not supported with any observable positive $X$ excursion. In Model 3 such $Z$ values may result in deviations of electrojet borders. Beyond the limits of 1D model such $Z$ excursions may be attributed to coastal effects or some vortice-like 2D structures. Potentially smaller confidence in $Z$ can be accounted for in the model (Eq. 6) attributing smaller weight to residuals in $Z$, e.g., with the coefficient 0.5. However, such an approach needs further statistical justification.

Usage of $Z$ is inevitable in our case, when number of stations is small. In Fig. 7 we illustrate the alternative reduced Model 3 run for the event of Figure 6, which does not take into account $Z$ component. The susbtantial difference appears only at 18–19 UT during the substorm expansion phase, when the reduced model reports much narrower electrojet with higher current density. Definitely a 2–3$^o$ wide electrojet in such condition is unphysical. Investigation of Fig. 7e shows, that proper knowledge of $Z$ is essential to calculate proper electrojet lcoation.

Finally, we are solving the considered mathematical problem with very generic maximum likelyhood approach, which allows priors, regularization, comprehensive error-handling, etc. This approach can be used in variety of other empiric model studies.

## 6   Conclusions

In this study, we investigated the models of westward auroral electrojet using magnetic field observations along a meridian chain of ground-based magnetometers. The model with one current strip and some corrective actions works reasonably well, when

the number of stations is small. Special attention needs to be taken in future to reconcile $X$ and $Z$ magnetic components.



## Appendix A: Auroral oval boundaries

5    Starkov (1994) model is actually an original Holzworth and Meng (1975) model of discrete and diffuse oval boundaries, but it uses $AL$ index instead of obsolete $Q$ index as input parameter. In our study we use only discrete aurora boundaries.

$$\theta = A_0 + A_1 \cos[15(t + \alpha_1)] + A_2 \cos[15(2t + \alpha_2)] + A_3 \cos[15(3t + \alpha_3)] \tag{A1}$$

where $\theta$ is boundary colatitude in corrected geomagnetic coordinates, $A_i$ are constants in degrees, $t$ is magnetic local time in hours, $\alpha_i$ are constants in hours. Constants $A_i$, $\alpha_i$ are determined separately for each boundary with respect to $AL$ index:

10   $$\left\{ \begin{array}{c} A_i \\ \alpha_i \end{array} \right\} = a_0 + a_1 \lg |AL| + a_2 \lg^2 |AL| + a_3 \lg^3 |AL|. \tag{A2}$$

Regression coefficients are in Table.





**Table A1.** Regression soefficients

|  | $A_0$ | $A_1$ | $\alpha_1$ | $A_2$ | $\alpha_2$ | $A_3$ | $\alpha_3$ |
|---|---|---|---|---|---|---|---|
| Polar boundary | | | | | | | |
| $a_0$ | $-0,07$ | $-10,06$ | $-6,61$ | $-4,44$ | $6,37$ | $-3,77$ | $-4,48$ |
| $a_1$ | $24,54$ | $19,83$ | $10,17$ | $7,47$ | $-1,10$ | $7,90$ | $10,16$ |
| $a_2$ | $-12,53$ | $-9,33$ | $-5,80$ | $-3,01$ | $0,34$ | $-4,73$ | $-5,87$ |
| $a_3$ | $2,15$ | $1,24$ | $1,19$ | $0,25$ | $-0,38$ | $0,91$ | $0,98$ |
| Equatorial boundary of auroral oval | | | | | | | |
| $a_0$ | $1,61$ | $-9,59$ | $-2,22$ | $-12,07$ | $-23,98$ | $-6,56$ | $-20,07$ |
| $a_1$ | $23,21$ | $17,78$ | $1,50$ | $17,49$ | $42,79$ | $11,44$ | $36,67$ |
| $a_2$ | $-10,97$ | $-7,20$ | $-0,58$ | $-7,96$ | $-26,96$ | $-6,73$ | $-20,24$ |
| $a_3$ | $2,03$ | $0,96$ | $0,08$ | $1,15$ | $5,56$ | $1,31$ | $5,11$ |
| Equatorial boundary of diffuse oval | | | | | | | |
| $a_0$ | $3,44$ | $-2,41$ | $-1,68$ | $-0,74$ | $8,69$ | $-2,12$ | $8,61$ |
| $a_1$ | $29,77$ | $7,89$ | $-2,48$ | $3,94$ | $-20,73$ | $3,24$ | $-5,34$ |
| $a_2$ | $-16,38$ | $-4,32$ | $1,58$ | $-3,09$ | $13,03$ | $-1,67$ | $-1,36$ |
| $a_3$ | $3,35$ | $0,87$ | $-0,28$ | $0,72$ | $-2,14$ | $0,31$ | $0,76$ |

## Appendix B: Electrojet current density estimate

Kamide et al. (1982) suggested the following estimate of the ionospheric east-west current density:

$$j_K(A \cdot km^{-1}) = \frac{2}{3} \times 1 \times \frac{10}{2\pi} H(nT), \tag{B1}$$

It is valid for the infinite equivalent ionospheric current approximation, assuming contribution from the ionospheric current to
5 the observed magnetic perturbation is twice that of the induction current flowing in Earth (similar to Eq. 4).

*Author contributions.* MAE performed the data processing. AAP is responsible for data analysis and interpretation.

*Competing interests.* The authors declare that they have no conflict of interest.



*Acknowledgements.* The data analysis was funded with Russian Science Fund project 18-47-05001. We are thankful for IMAGE data archive
and Prof. A.N.Zaitsev for Yamal data.





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





**Figure 6.** Model 3 for Example 2. (Left): measured and magnetic time profiles, model current density (standard deviation range is given with thin curves), electrojet boundaries (black — Starkov (1994) model, blue and red — Model 3). Vertical lines denote time instants for the right panel. (Right): Latitude cuts with model parameters for four time instants. Error bars in panel d show standard deviations. Magenta model in Fig. 6d corresponds to the variant with magenta dashed line in Fig.6e,f. Measured field is shown with stars, model field — with open circles.





**Figure 7.** Variants of Model 3 for Example 2. (Left): model current density and electrojet boundaries (black — full Model 3 with $X$ and $Z$ inputs, red — reduced Model 3 with only $X$ input). (Right): Latitude cut for 1730:30 UT for two model variants and also for a variant with current density fixed with Kamide et al. (1982) estimate. Measured field is shown with black stars, model field and current — with open circles. Error bars in panel c show standard deviations.