# Peer review of "Estimation of westward auroral electrojet current with sparse magnetometer chain data"

_Annales Geophysicae, 2019_

## Referee Comment (RC1) · Vladimir Papitashvili (Referee) · 7 Aug 2019

Referee's review of the following paper: Journal: Annales Geophysicae (ANGEO) Title: Estimation of westward auroral electrojet current with magnetometer chain data Author(s): Marina A. Evdokimova and Anatoli A. Petrukovich MS No.: angeo-2019-96 MS Type: Regular paper

The reviewed paper presents a new approach to the interpretation of ground-magnetometer data to reconstruct ionospheric currents that is applicable to a wide range of applications over the polar, mid-latitude, and equatorial region. The new modeling approach takes care of several issues that plagued previous models such as

false minima and when the number of magnetometers along a geomagnetic meridian is limited.

The presentation of the methodology, model description, and example results is clear and up to international standards. The length of the paper is adequate and the language id fluent and precise.

The Authors reached substantial conclusions with their new algorithm for reconstruction of ionospheric currents. The title and abstract are pertinent and understandable.

The mathematical justification of the new modeling approach is very clear, figures are of excellent quality and good size. The Authors give full credits to related works and indicate clearly their own contributions.

The scientific value of this paper is important for more automated reconstruction of ionospheric currents from existing and future magnetometer networks and arrays. I recommend publishing this paper in Annales Geophysicae.

---

## Referee Comment (RC2) · Anonymous Referee #2 · 1 Oct 2019

The study describes new alternatives to dealing with modeling issues related to the limited coverage of data-points and false minima of the optimized functions. Even though the manuscript addresses models of the westward auroral electrojet, their claims are useful for the study of electric currents at other magnetic latitudes.

In general, the manuscript is well structured, has an adequate length, a good mathematical description, and a fluent language, though it has some typos and grammatical issues. The title is okay, although it could be more precise. Perhaps the authors want to strength the fact that the study focuses on a comparison between two models and a description of new assumptions to get better results.

At first glance, it appears that this study should be published in Annales Geophysicae. However, there are some minor points that the authors must address before publication, which might help to illustrate better their results to the readers, including a proper conclusion section. Comments are as follows,

- It is a bit disappointing to find a three-line conclusion section after going through such an interesting paper. Please, profit from this section to make your claims and findings clear and concise.

- Although in general, the authors give credits to related works, the readers might profit from the references of the following statements:

Page 1, line 21. "In the course of a substorm, the activity zone first shifts equatorwards during growth phase, then, after an onset, it retreats polewards. For stronger substorms auroral zone shifts equatorwards."

Page 9, line 32. "Current density and width of the electroject are strongly anticorrelated"

- It could be more comfortable to picture the magnetometer arrangement by showing a map of the stations rather than a table with the coordinates.

- For all the figures, a more detailed description in the caption is very much appreciated.

- Figure 5 is not described throughout the manuscript. Please delete it or use it.

- In the caption of Figure 5, the authors mention Figure 5 instead of Figure 4. It is difficult to distinguish among the circles and dots (black circle together with a black dot make a bigger black dot). Please use different forms (e.g., +, *).

- The combination of lines in red and magenta in figures 3 and 6 are hard to follow. Maybe the authors want to use contrasting colors.

- In Figure 6, please do not use dashed-lines. When the time series is highly variable, it generates noise. Please use solid lines with different colors.

- Finally, I encourage the authors to have a closer look at the language. As mentioned before, there are several typos and some grammatical issues.

---

## Author Comment (AC1) · 6 Nov 2019

**Referee #1**

We thank the referee for attentive reading and overall assessment.

**Anonymous Referee #2**
We would like to thank the referee for useful advices and comments

*The study describes new alternatives to dealing with modeling issues related to the limited coverage of data-points and false minima of the optimized functions. Even though the manuscript addresses models of the westward auroral electrojet, their claims are useful for the study of electric currents at other magnetic latitudes.*
*In general, the manuscript is well structured, has an adequate length, a good mathematical description, and a fluent language, though it has some typos and grammatical issues. The title is okay, although it could be more precise. Perhaps the authors want to strength the fact that the study focuses on a comparison between two models and a description of new assumptions to get better results.*

We suggest a small modification of the title, making it more specific:

Estimation of westward auroral electrojet current with **sparse** magnetometer chain data

*At first glance, it appears that this study should be published in Annales Geophysicae. However, there are some minor points that the authors must address before publication, which might help to illustrate better their results to the readers, including a proper conclusion section. Comments are as follows,*
*- It is a bit disappointing to find a three-line conclusion section after going through such an interesting paper. Please, profit from this section to make your claims and findings clear and concise.*

The conclusion section has been expanded. We don't think that it is necessary to list all results here, the approach is summarized in a special section. Here we intend to mention mostly outstanding problems.

The new text is

In this study, we investigated the models of westward auroral electrojet using magnetic field observations of sparse meridian chains of ground-based magnetometers. The model with one current strip works reasonably well, even using only three stations and two magnetic field components X and Z. Some corrective actions proved to be necessary to avoid general computational problems related with unphysical minima in the nonlinear optimization algorithm. However, the model naturally cannot reliably estimate location of electrojet boundary in a case of lack of stations near that boundary. Special attention also needs to be given in future to reconciliate sometimes contradictory profiles of X and Z magnetic components.

*- Although in general, the authors give credits to related works, the readers might profit from the references of the following statements:*
*Page 1, line 21. "In the course of a substorm, the activity zone first shifts equatorwards during growth phase, then, after an onset, it retreats polewards. For stronger substorms auroral zone shifts equatorwards."*

The reference has been added (Akasofu, 1968).

*Page 9, line 32. "Current density and width of the electrojet are strongly anticorrelated"*

This is not an independent statement. It is specific feature of our model, supported with the covariance matrix analysis (in the parameter space). We add more description in the relevant place

The new text:
Current density and width of the electrojet are strongly anticorrelated in the model **with one strip and two-three magnetic stations**. Almost the same magnetic field can be produced be a variety of strips with the different widths and current density, but the same total current.

*- It could be more comfortable to picture the magnetometer arrangement by showing a map of the stations rather than a table with the coordinates.*

We add map of Yamal stations and give a reference to IMAGE picture because it is rather famous. So, this map became Figure 1 and other figures numbers are shifted.

[Figure]

*- For all the figures, a more detailed description in the caption is very much appreciated.*
*- Figure 5 is not described throughout the manuscript. Please delete it or use it.*

The description of Figure 6 (now it is Figure 7) is expanded.
The reference on this Figure (now it is Figure 6) was added in the subsection 3.3.

The new text:
The second approach is to use a so-called multistart algorithm. We generate a normally randomized set of initial conditions around a Starkov model solution, run Model 3 several times, and choose a result with the minimal residuals (Eq. 6). For the case of Fig. 5 (right panel, blue curve) we show the map of 50 initial conditions (for the boundary locations only) on the Fig.6. Starkov model is shown with red point. The solutions, starting from the filled black circles lead to the absolute minimum, shown by filled blue point. The empty black circles lead to the local

minima (blue open circles). Since Model 3 is computationally simple, the method works well, and it is not necessary to fill densely the parameter space during the randomization.

*- In the caption of Figure 5, the authors mention Figure 5 instead of Figure 4. It is difficult to distinguish among the circles and dots (black circle together with a black dot make a bigger black dot). Please use different forms (e.g., +, *).*

We compared the initial variant and the new one below. We think the initial figure is easier for perception.

[Figure]

[Figure]

*- The combination of lines in red and magenta in figures 3 and 6 are hard to follow. Maybe the authors want to use contrasting colors.*
*- In Figure 6, please do not use dashed-lines. When the time series is highly variable, it generates noise. Please use solid lines with different colors.*

Yes, it makes sense. We tried to satisfy that comments. Both figures are below. Magenta was replaced with green on Figure 3 (now it is Figure 4) and magenta dashed and solid lines - with

green and orange solid lines on Figure 6 (now it is Figure 7). Also we deleted circles on the field-latitude plots as excess elements.

[Figure]

[Figure]

- *Finally, I encourage the authors to have a closer look at the language. As mentioned before, there are several typos and some grammatical issues.*

We carefully rechecked the text. Also the copyediting service helps a lot with the simple corrections.